# The 3.0 Cell Communication: New Insights in the Usefulness of Tunneling Nanotubes for Glioblastoma Treatment

**DOI:** 10.3390/cancers13164001

**Published:** 2021-08-08

**Authors:** Lorenzo Taiarol, Beatrice Formicola, Stefano Fagioli, Giulia Sierri, Alessia D’Aloia, Marcelo Kravicz, Antonio Renda, Francesca Viale, Roberta Dal Magro, Michela Ceriani, Francesca Re

**Affiliations:** 1BioNanoMedicine Center NANOMIB, School of Medicine and Surgery, University of Milano-Bicocca, 20854 Monza, Italy; l.taiarol@campus.unimib.it (L.T.); beatrice.formicola@unimib.it (B.F.); s.fagioli@campus.unimib.it (S.F.); g.sierri@campus.unimib.it (G.S.); marcelo.kravicz@unimib.it (M.K.); a.renda3@campus.unimib.it (A.R.); f.viale1@campus.unimib.it (F.V.); roberta.dalmagro@unimib.it (R.D.M.); 2Department of Biotechnology and Biosciences, University of Milano-Bicocca, 20126 Milan, Italy; alessia.daloia@unimib.it (A.D.); michela.ceriani@unimib.it (M.C.)

**Keywords:** tunneling nanotubes, glioblastoma, stem cells, tumor microenvironment, drug delivery, nanoparticles

## Abstract

**Simple Summary:**

Communication between cells helps tumors acquire resistance to chemotherapy and makes the struggle against cancer more challenging. Tunneling nanotubes (TNTs) are long channels able to connect both nearby and distant cells, contributing to a more malignant phenotype. This finding might be useful in designing novel strategies of drug delivery exploiting these systems of connection. This would be particularly important to reach tumor niches, where glioblastoma stem cells proliferate and provoke immune escape, thereby increasing metastatic potential and tumor recurrence a few months after surgical resection of the primary mass. Along with the direct inhibition of TNT formation, TNT analysis, and targeting strategies might be useful in providing innovative tools for the treatment of this tumor.

**Abstract:**

Glioblastoma (GBM) is a particularly challenging brain tumor characterized by a heterogeneous, complex, and multicellular microenvironment, which represents a strategic network for treatment escape. Furthermore, the presence of GBM stem cells (GSCs) seems to contribute to GBM recurrence after surgery, and chemo- and/or radiotherapy. In this context, intercellular communication modalities play key roles in driving GBM therapy resistance. The presence of tunneling nanotubes (TNTs), long membranous open-ended channels connecting distant cells, has been observed in several types of cancer, where they emerge to steer a more malignant phenotype. Here, we discuss the current knowledge about the formation of TNTs between different cellular types in the GBM microenvironment and their potential role in tumor progression and recurrence. Particularly, we highlight two prospective strategies targeting TNTs as possible therapeutics: (i) the inhibition of TNT formation and (ii) a boost in drug delivery between cells through these channels. The latter may require future studies to design drug delivery systems that are exchangeable through TNTs, thus allowing for access to distant tumor niches that are involved in tumor immune escape, maintenance of GSC plasticity, and increases in metastatic potential.

## 1. Introduction

Glioma is the most common primary tumor of the central nervous system, with an annual incidence of approximately 6 cases per 100,000 individuals worldwide and with approximately 50% of them being classified as glioblastoma (GBM). GBM is the most aggressive form of glioma, with a median lifespan from time of diagnosis to death of approximately 15 months.

Based on its histological appearance, GBM has been traditionally classified as an astrocytoma, though the precise cell type from which the disease originates is still a controversial issue. Some experts argue that the GBM origin is a subpopulation of neural stem cells, while others claim that it derives from the transformation of differentiated astrocytes [1]. Regardless, little progress has been made in GBM therapy, with no change in the standard of care for almost 20 years [2]. The current therapeutic approach for newly diagnosed GBM patients is based on surgery, followed by temozolomide chemotherapy and radiotherapy in combination with corticosteroids. In addition, in 2015, a noninvasive technique based on the application of alternating electrical fields (tumor treating fields, TTF) was approved as an adjuvant therapy for newly diagnosed GBM [3]. GBM is assigned WHO grade IV [4], and recently, the classification has been refined, with diagnosis based not only on histology but also on several molecular markers such as isocitrate dehydrogenase (IDH) and epidermal growth factor receptor (EGFR) [5]. Unlike other cancers, GBM remains confined in the brain without any systemic spread [6]. However, almost every GBM recurs, and recurrent tumors are chemotherapy-resistant, with higher invasiveness and aggressiveness compared with the original tumor [7,8,9]. Consequently, there is no standard treatment for recurrent GBM, partially due to poor biological knowledge of the disease [10,11]. Recent studies on tumor heterogeneity suggest that residual tumor cells after whole total tumor resection share only 60–80% of their mutations with the primary tumor and differ significantly in terms of gene expression profile, microenvironment, and extent of immune cell infiltration [12,13]. Additionally, cancer stem cells play a pivotal role in GBM recurrence, though there is no generally accepted definition of them within GBM and how they specifically contribute to therapy resistance and tumor recurrence has not been clarified.

This review describes the main data currently available about the communication modalities between the different cells composing the GBM environment, focusing on tunneling nanotubes (TNTs), which are described in detail. Finally, an overview of potential therapeutic approaches based on TNTs is presented.

## 2. Main Cell Types Interacting in Glioblastoma Microenvironment

The GBM mass consists not only of a heterogeneous population of cancer cells but also of a variety of resident and infiltrating host cells, secreted factors, and extracellular matrix components, which collectively create the tumor microenvironment (TME). The TME is a complex network of signals and trafficking able to regulate tumor growth and invasiveness, angiogenesis, and chemotherapy resistance and to modulate immune response, drug delivery, and therapeutic responses [14,15]. Consequently, innovative therapeutic approaches should be designed considering the GBM microenvironment. However, there is currently limited understanding of cell communication between tumor and non-tumor cell types in the TME. In this section, we provide an overview about the main cells composing the GBM TME, which can communicate between them.

### 2.1. Tumor Immune Cells

Increasing evidence has revealed that infiltrating immune cells and other stromal components in the TME, of which the proportions vary according to cancer stage, are associated with the prognosis of GBM. Maintenance of the TME is one of the crucial factors influencing local immune dysfunction, which plays a critical role in GBM-induced immunosuppression [6]. This milieu in turn leads to immunotherapeutic treatment failure [1]. The immunosuppressive TME is mainly caused by recruited immunosuppressive cells [13], tumor-derived immunosuppressive factors [14], overexpressed immune checkpoints [16], and GBM cell epigenetics that silence HLA molecules [17].

Microglia cells and tumor-associated macrophages (TAMs) are tissue-resident cells (comprising 15% of the TME) and bone marrow-derived macrophages (comprising 85% of the TME) that may regulate tumorigenesis and are collectively known as glioblastoma-associated microglia and macrophages (GAMs) [18]. There are two main TAMs subtypes, known as M1 and M2. The former is capable of antitumor activity, promoting cytotoxic and inflammatory effects; on the contrary, the latter subtype is pro-tumoral, with its anti-inflammatory activity switching off the host immune response against GBM. These cell types are considered important targets for GBM therapy [19].

Along with TAMs, CD4^+^T and CD8^+^T cells are capable of influencing tumorigenesis by receiving inhibitory signals from other TME cells and cancer cells, which lead to immune exhaustion and tumor tolerance [20].

A major mechanism leading to tumor immune-tolerance is activation of the PD-1/PD-L1 pathway. GBM cells massively express programmed cell death-ligand 1 (PD-L1) and consequently bind more programmed cell death protein 1 (PD-1) on T cells. This ligand–receptor interaction inhibits the differentiation of T cells into T effectors and promotes the switch towards other phenotypes such as T regulatory (T_regs_) and T exhausted cells. This phenomenon prevents the tumor rejection usually mediated by CD8^+^ lymphocytes [21,22,23]. T_regs_ are potent immunosuppressive cells that cause GBM immune escape. T_regs_ do not exist in normal human brain tissue, but a large number of immunosuppressive T_regs_ have been found in the GBM TME [24].

It has been also suggested that, in brain tumors, dendritic cells (DCs) recognize and present tumor-derived antigens inside the brain tissue or in the draining lymphoid stations in order to boost a T effector cell response against cancer cells [25,26]. These cells are normally not present in the healthy brain parenchyma. However, during tumorigenesis, they can reach the brain tissue via afferent lymphatic vessels and/or endothelial venules [27,28,29]. Drainage of tumor antigens into cervical lymph nodes has been observed in animal models via the glymphatic system [30,31]. The glymphatic system is a functional meningeal system located in the dura mater, which allows for the passage of molecules and immune cells into the deep cervical lymph nodes [32,33,34], where internal recirculation mechanisms involve the cerebrospinal fluid and interstitial fluid [16]. Moreover, cervical lymph nodes may have the property to modulate the immune response to tumor antigens toward either tolerance or reactivity [35]. Even if the specific role of DCs in the GBM environment is not yet elucidated, it is accepted that they have a pivotal role in antitumor immunity [36,37,38].

### 2.2. Glioblastoma Stem Cells

Several subtypes of GBM have been defined on the basis of different molecular alterations and gene expression patterns. Verhaak et al. described four GBM molecular subtypes: classical, proneural, mesenchymal, and neural [39]. These subtypes might arise from multiple stem cell-like populations through distinct differentiation pathways.

GBM stem cells (GSCs) are defined as a quiescent subpopulation of cancer cells that have high self-renewing abilities, clonal tumor initiation capacity, and long-term repopulation potential [40]. In addition, the contribution of GSCs to GBM malignancy is widely accepted, as these cells can recreate the tumor mass after surgery. GSCs can be classified as proneural (PN) or mesenchymal (MES) subtypes because of their gene expression profiles and distinct biological characteristics [41]. GSCs are located in different niches, specific protective TME regions in GBM tumors where they can preferentially interact with specific cell types. For example, GSCs in the perivascular niche can mutually communicate with endothelial cells that secrete soluble cues, thus supporting the GSCs’ self-renewal. In return, GSCs release vascular endothelial growth factor (VEGF) and stromal-derived factor 1, thus promoting angiogenesis [42,43].

In addition, GSCs transdifferentiate into pericytes and contribute to the vascular structure [44,45]. Another phenomenon, called “vasculogenic mimicry”, takes place in GBM, where GSCs differentiate into endothelial-like cells, forming vessel-like structures. These structures are able to supply the tumor cells with nutrients and oxygen [46,47]. GSCs can also communicate with immune cells, promoting the establishment of a suppressive TME and thus allowing for tumor immune escape and progression [44]. In return, GAMs promote GSCs metabolic pathways to gain energy [48]. Furthermore, GSCs directly regulate immune cells, leading to the activation of T_regs_, the inhibition of cytotoxic T cell proliferation, and the induction of cytotoxic T cell apoptosis [40,49]. In summary, GSCs present in these niches preserve their phenotypic plasticity, protect themselves from the immune system, facilitate GBM metastasis, and are resistant to commonly employed cancer therapies. This is one of the main reasons why targeted therapy has not demonstrated efficacy in phase 3 clinical trials against GBM so far.

### 2.3. Other Brain Cells

GBMs are among the most vascularized solid tumors found in humans, and blood vessels play a key role in supporting tumor progression [50], especially the endothelial cells (ECs) that are closely associated with GSCs [45]. Therefore, hindering communication between the endothelium and GSCs might represent a strategy to hamper GSCs survival. GSCs proliferation and invasiveness are also supported by the presence of aberrant tumor-derived vasculature that is usually associated with a higher degree of malignancy and a poor prognosis. Thus, high vascularization together with the vasculogenic mimicry contribute to the failure of antiangiogenic therapies against GBM [51,52,53].

Astrocytes, essential components in the structure and function of the blood–brain barrier (BBB), have been shown to support tumor angiogenesis via multiple mechanisms including secretion of angiogenic and growth factors, such as VEGF, and protein carriers, such as insulin and albumin. Astrocytes surrounding GBM commonly undergo functional and phenotypical changes through astrogliosis, a process in which reactive astrocytes secrete a large number of soluble factors that promote GBM invasiveness, proliferation, and migration [54]. In turn, tumor cells suppress p53 expression in astrocytes, thus promoting GBM cell survival through modulation of the extracellular matrix composition [55,56].

In addition, neuronal‒GBM cell interactions have recently emerged as an important factor in tumor growth, as neuronal activity promotes tumor progression by inducing the release of brain-derived neurotrophic factor (BDNF) [57], neuroligin-3 [58], and dopamine [59]. Furthermore, GBM cells can increase local neuronal excitability, a mechanism that is responsible for the presence of seizures in GBM patients. In addition, this last event fosters the excitability-dependent secretion of mitogens, generating a vicious cycle to sustain the GBM microenvironment [60].

## 3. Cell Communication Modalities

The high complexity of TME aroused huge scientific interest about the communication modalities between cells in GBM. Understanding this communication could open new avenues for therapy design. The GBM progression and invasion of the brain involves multiple communication strategies between the cells that compose the TMEs, for instance, secretion of soluble factors such as chemokines and cytokines, extracellular vesicles (EVs; including exosomes and microvesicles), and direct cell–cell contact (gap junctions, nanotubes, and microtubes) [56].

Gap junctions are involved in the exchange and transfer of small molecules such as Ca^2+^, ATP, and metabolites between two adjacent cells [61]. These clusters of intercellular channels are composed of connexins, the core proteins in gap junctions, in which connexin43 (Cx43) is expressed by human cerebral microvascular ECs (hCMEC), astrocytes, and GBM cells. High levels of this protein enable the formation of multicellular networks and allow cell-to-cell communication via calcium waves.

An increase in Cx43 expression in GBM-initiating cells has been associated with increased invasiveness [61,62,63]. Connexins may also increase metastatic cell growth in GBM and resistance to standard therapies [64,65,66]. Despite the evidence of connexins role in malignant conditions, gap junctions are fundamental in tissue homeostasis, regulation of cell growth, and differentiation, making their targeting challenging without inducing toxic side effects [67].

EVs are membrane-bound vesicles secreted by all cells into the extracellular space in both physiological and pathological conditions, thus representing a powerful strategy for intercellular communication. Accordingly to their biogenesis, size, release pathways, cargo, and function, EVs can be divided into a wide range of vesicle types including exosomes (50–200 nm), microvesicles (>100 nm–1 µm), apoptotic bodies (50 nm–2 µm), and large oncosomes (>1 µm) [68,69]. EVs are exploited by cells for the transport of transcription factors, lipids, proteins, enzymes, and several metabolites, thus influencing the physiology and the phenotype of the receiving cell. The presence of EVs in almost every tissue, brain included, opens an entirely new perspective on cellular communication in GBM. Cancer cells usually exhibit higher EVs release compared with their healthy counterparts. Moreover, EVs belonging to glioma cells contain altered and different cargos than normal glial cells, resulting in the transfer of oncogenic activity. In the context of GBM, tumor-derived EVs are able to change the phenotype of normal cells to promote angiogenesis, tumor cell invasion, immune suppression, and altered metabolic regulation. Furthermore, EVs are able to switch to a normal cell phenotype and to promote angiogenesis, tumor cells invasion, immune suppression, and altered metabolism [70,71,72].

Pace K.R. et al. reported that exosomes decorated with L1 protein, which are normally involved in neuronal development, are upregulated in many types of cancers, such as GBM. This increase may affect proliferation and cell motility and may promote tumor invasiveness in vitro [73]. Similarly, Lane R. et al. suggested that GBM subtypes could be distinguished based on EVs cargo, thus proposing the existence of subtype-specific EVs-associated biomarkers that are also involved in the regulation of tumor aggressiveness and recurrence. The diagnostic identification of such biomarkers would be helpful to develop personalized treatments for GBM patients [74]. Moreover, EVs are also involved in the recruitment, activation, or suppression of an innate immune system, particularly by mediating immunosuppressing mechanisms such as the induction of immunosuppressive monocytes without directly inhibiting T cell PD-L1 expression [75].

## 4. Tunneling Nanotubes

The communication mechanisms between cells in a tumor matrix remain poorly understood, especially for cells that are distant from each other. In the last two decades, experimental evidence from different research groups proved the existence of thin membranous tubes that interconnect cells called tunneling nanotubes (TNTs), tumor microtubes (TMs), or membrane bridges.

### 4.1. TNT Discovery, Characteristics, and Classification

TNTs are dynamic connections between two or more cells, which act as a route for cell-to-cell communication. Discovered in PC12 cells by Rustom et al. in 2004, TNTs have been defined as open-ended channels mediating membrane continuity between two or more cells over short to long distances [76]. TNTs are heterogeneous, transient, highly dynamic and sensitive nanotubular structures that connect cells by creating a complex network [76]. Unlike filopodia, TNTs are not branched and are suspended above the matrix [77].

Connecting distances up to 120 μm, TNTs can be classified as two types: (1) short (100–200 nm), thin (≤0.7 μm), and dynamic type I nanotubes are made of actin and formed by cells within their surroundings to make cell–cell contact; (2) long (1 μm), thick (≥0.7 μm), and more stable type II nanotubes are made of tubulin and cytokeratin filaments and formed by the detachment of two cells that are already connected [78,79,80] (Figure 1). Type II TNTs transfer cellular cargo between neighboring cells. Interestingly, it has been shown that different classes of TNTs exist even within a single cell type [81,82].

TNTs can have different lifetimes ranging from a few minutes (T cells) to hours (PC12 and NRK cells). These differences may reflect the existence of different subclasses of TNT-like structures supporting the experimental heterogeneity seen in TNT visualization [77].

These long membranous nanotubular structures have been identified in vitro in diverse cell types including neuronal cells, epithelial cells, almost all immune cells, and tumor cells [76,83,84]. In has been recently shown that neuronal-like cells are able to form TNT-like structures, which are implicated in the cell-to-cell spreading of Tau aggregates, thereby worsening the progression of Alzheimer’s disease and other tauopathies [85]. Multiple studies have shown the capability of some epithelial and mesenchymal cancer cell types to form very long TNTs and TMs that are involved not only in the increase in malignancy but also in the formation of the tumor ecosystem connecting different cells types between them [86,87].

Although clearly demonstrated in vitro, the presence of TNTs in mammalian tissues in vivo has initially been questioned because of technical limits that made their detection arduous. The identification of TNTs in the mammalian cornea has provided the first evidence of membrane nanotubes in vivo [88], opening the possibility to removing the word “like” from the structures observed in vivo, which are currently carefully indicated as “TNT-like structures” [89,90,91].

### 4.2. TNTs Formation

TNTs are transient structures that can originate by two mechanisms: de novo or via cell dislodgments [92]. De novo generation is an actin driven process whereby filopodia-like protrusions elongate via actin polymerization, probably initiated by Rho GTPases, until they reach a target cell. The resulting physical contact then results in membrane fusion [76].

The cell dislodgement mechanism occurs when two migrating cells that are initially in contact separate, thereby generating a nanotube. This mechanism has been observed in T cells, macrophages, and natural killer cells [81]. In this process, the time of contact is essential; a transient contact of less than 4 min hardly gives rise to TNTs [93]. Studies have mainly focused on well-known molecular machinery involved in blocked/reduced/activated TNT formation, such as filamentous actin (F-Actin) [94], M-Sec, RalA GTPase, LST1, myosin Va and X [76], and Cx43 [89] and plays a key role in defining the biology, the mechanisms, and the forces involved in TNT dynamics. Moreover, I-Bar lipid raft proteins [95] are required to generate membrane curvature during TNT formation as adhesion molecules such as N-cadherin and β–catenin [96] are fundamental in TNT guidance and initiation.

Several research groups have demonstrated the presence of connexin and gap junction channels in TNTs, opening a long-range gap junctional communication mediated by the TNTs processes [97], probably because the two communication systems evolved to complement each other in a coordinating cell-to-cell communication. Different cancer cell models exhibit spontaneous formation of TNTs when cultured in vitro [76,98,99,100], while epithelial HBEC-3 cells, which are non-tumorigenic, rarely form TNTs when cultured in vitro [98]. Contrarily, the exposure of cell cultures to stress conditions can induce the formation of TNTs. In 2011, Wang et al. demonstrated that H_2_O_2_ exposure or starvation induced by serum depletion can produce TNTs in rat hippocampal astrocytes and neurons [101]. Moreover, TNT formation correlates with different stress factors such as infection [102], inflammation [88,103], hypoxia [104], ultraviolet exposure [105], X-ray exposure [106], and particle radiation exposure [107]. The role of TNTs after irradiation is poorly characterized. However, in 2015, Osswald et al. showed the protecting effect of TNTs after X-ray radiation, while in 2019, Reindl et al. demonstrated a decrease in connections after α-particle irradiation.

### 4.3. TNT Function, Biological Effects, and Mechanisms of Transport

Evidence indicates that TNTs play a role in intercellular exchanges of signal clues, molecules, organelles, and pathogens, implicating them in a diverse array of physiological functions and pathological events. In particular, TNTs allow for the exchange of mitochondria [108], lysosomes [109], vesicles [110], proteins [111], viruses [112,113], and miRNAs [114] between connected cells.

Furthermore, multiple studies have described the role of TNTs in the intercellular exchange of Ca^2+^ signals between distant cells, suggesting a continuity of membrane and cytosol of TNTs, and active gap junction channels [115].

Different communication systems may coordinate, interact, and develop to complement each other, as demonstrated by long-range transmission of inositol triphosphate mediated by TNTs through a gap junction-dependent mechanism [79,115].

Armulik et al. have described the communication between pericytes exploiting TNTs, thus confirming the essential role of these branched cells in regulation and brain homeostasis [116]. Triple immunostaining confocal microscopy analysis revealed pericytes as the main source of TNTs during neuro-angiogenesis, both in the early phases in physiological conditions and in tumor state [79].

Other than exerting physiological functions, TNTs are involved in cancer development, in reprogramming of malignant cells, and in alterations of the TME. In certain cases, TNTs can promote invasiveness and protection of cancer cells from cytotoxic drugs. It has been shown that the mitochondrial transfer between leukemic cells via TNTs increases chemotherapy resistance [117].

Desir s. et al. have demonstrated that in colon cancer cells, TNTs transfer KRAS oncogene, resulting in a heterogeneous distribution of mutant KRAS that profoundly modulates the TME and subsequent tumor progression [118].

Similarly, diffusion of the DNA repair enzyme O^6^-alkylguanine DNA alkyltransferase (MGMT) through TNTs mediates the protection of GBM cells against temozolomide [67]. In other conditions, TNTs may promote drug distribution between target cells, highlighting their potential beneficial properties for therapy [119].

An intriguing note to clarify is the mechanism of cargo transport that could be unidirectional. It may be, for instance, that actin and myosin drive transport, in which cellular components are anchored and driven by the directional actin polymerization at one end. Alternatively, it may be bidirectional from TNTs containing microtubules, in which cargo is moved by microtubule-based molecular motors (kinesin/dynein-mechanism) [120]. In both mechanisms, ATP is required, as they are active transports [81]. Another active cytoskeleton-independent transport (vesicular dilatation or “gondolas”) is present in both type I and type II TNTs. The two mechanisms of transport differ in dilatation along the tubes, speed, and direction [78].

## 5. Tunneling Nanotubes in Glioblastoma

### 5.1. Formation of TNTs in GBM

TNTs have been reported in several in vitro, ex vivo, and in vivo tumor models of GBM. The presence of TNTs have been demonstrated and characterized in vitro in cultures of C6 glioma cells, U87MG cells, U251 cells, primary human-derived brain tumor cells, and patient-derived GBM stem cells [15,121].

In terms of functional analysis, Civita et al. demonstrated the trafficking of mitochondria from astrocytes to GBM cells via F-actin TNT structures using both 2D and 3D in vitro GBM co-culture models, indicating that contact communication between non-neoplastic astrocytes and tumor cells may occur [122]. Interestingly, we recently showed that TNTs formed by healthy astrocytes or by GBM cells display structural differences in terms of length and thickness, which reflect different transport efficiencies [123]. A recent paper showed that irradiated U87 GBM cells quickly establish a network of cell-to-cell connections with high TNT content in comparison with non-irradiated cells, suggesting that the TNT formation may be also a consequence of treatment [124].

In vivo, in a syngeneic astrocytoma mouse model, it has been shown that many tumor cells extend ultra-long membrane protrusions and use TMs and TNTs as routes for brain invasion, proliferation, and interconnection over long distances [106].

In general, cancer cells exploit this physical connection to exchange material between themselves or with the cells present in TME (Figure 2A). In vitro and in vivo studies published by Errede et al. also revealed the existence of brain pericyte-derived TNTs that appear to be involved in exploring the surrounding microenvironment, searching for and connecting with targeted vessels, and contributing to the pathological angiogenesis in GBM. In particular, the results revealed that TNTs are formed both among different pericytes and between pericytes and other brain cells, especially ECs.

Moreover, since GBM is a highly vascularized tumor [125], the exchange of molecular messages and/or organelles through pericytes and ECs by TNTs may contribute to the tumor spreading [79].

In terms of ex vivo work, Pinto et al. have shown for the first time that GSCs obtained from GBM patients were able to form TNTs and TMs in culture and were able to exchange organelles. These cells were obtained from the infiltrative tumor niche, which is responsible for GBM recurrence [121], suggesting that TNTs could be involved in long-range cell-to-cell communication in GBM (Figure 2B).

### 5.2. Role of TNTs in GBM

TNT cell-to-cell communication allows tumor cells to acquire new abilities, such as enhanced plasticity, migratory phenotypes, angiogenic ability, and therapy resistance, which can contribute to cancer aggressiveness, invasiveness, and recurrence [67,126].

In this context, Valdebenito et al. have shown that TNTs mediate the chemo- and radiotherapy resistance of GBM by exchanging MGMT protein from resistant to sensitive cells, suggesting a role of TNTs in promoting a more malignant phenotype [67].

Moreover, MGMT proteins diffuse long distances, suggesting that TNTs can be formed with GBM cells located far away from the primary tumor.

In co-culture experiments, astrocytes surrounding U87 GBM cells enhanced TNT formation toward tumor cells and exploited this physical connection to transfer undamaged mitochondria, useful molecules, or energy substrates to GBM cells in order to modulate cell behavior and response to cytotoxic agents [122].

A continuously growing body of evidence suggests that immune cells, both brain-resident and infiltrated from the periphery, play key roles in GBM progression and invasiveness.

To the best of our knowledge, there is only one study reporting that mast cells (MCs), perivascular immunomodulatory cells, may form functional TNTs in vitro to communicate among themselves and with U251 GBM cells. This communication allowed for the bidirectional transfer of mitochondria and secretory granules [127].

In addition, the same authors speculate that TNTs provide a way for MCs to “alert” other cell types with a specificity that is not present when the mediators are secreted into the tissue microenvironment.

Despite recent studies on TNTs in GBM, many aspects of their function, biology, and mechanism of formation are still poorly understood and the progression of science in this important field is useful in developing new therapeutic strategies in pathological conditions.

## 6. TNTs as a Novel Strategy to Enhance Tumor Drug Delivery

TNTs are involved in the multistep process of cancer development from tumorigenesis to treatment resistance. Therefore, the scientific community is committed to characterize these communication systems for therapeutic purposes. Two possible strategies could be pursued: the pharmacological inhibition of TNT formation (Figure 3A) or the exploitation of TNTs as drug delivery channels (Figure 3B).

### 6.1. Inhibition of TNTs Formation

The discovery of TNTs biomarkers could be useful to generate targeting molecules that effectively inhibit TNTs formation. Currently, the only two TNTs markers known are actin and M-Sec (also referred to as tumor-necrosis factor alpha-induced protein 2), which are, unfortunately, not specific for TNTs [128].

The pharmacological targeting of TNTs in vitro has been performed using several molecules (reviewed in [15]), from inhibitors of actin polymerization to inhibitors of DNA synthesis, with positive results in terms of inhibiting formation or trafficking along TNTs. However, the clinical relevance of this approach has yet to be determined. In 2016, Desir et al. evaluated metformin and everolimus, two drugs that have received FDA approval for GBM treatment as TNT formation inhibitors in ovarian cancer cells. The results revealed that these compounds were able to interfere with TNT development by acting on the mammalian target of rapamycin (mTOR) pathway [104]. These data are in agreement with previous results obtained by Lou et al. [86], where the TNT formation in human mesothelioma cells was inhibited by metformin and everolimus. However, in vivo preclinical data about the applicability of these mTOR inhibitors are not available yet.

### 6.2. Exploiting TNTs as Drug Delivery Channels

A more creative alternative would be the exploitation of these membranous cellular structures as a novel Trojan horse strategy to strengthen tumor drug delivery. Consequently, the exchange of drug delivery systems between cells composing the TME via TNTs comes as an interesting and advantageous opportunity to reach tumor cells, which usually escape current therapies. Moreover, exploiting the capability of TNTs to make physical contact with even distant cells indicates that they could be used to reach metastatic cells located in different tumor niches.

In the context of GBM therapy, different chemotherapeutics have been encapsulated in nanosystems (e.g., nanoparticles, Box 1) to facilitate BBB crossing and/or to enhance their brain penetration and tumor targeting, thereby reducing side effects and improving drug concentration in the TME. A plethora of multifunctional nanoparticles have been synthetized and characterized as potential devices for GBM therapy (reviewed in [23,129]). Among them, liposomes and polymeric nanoparticles are the most promising in terms of versatility and biocompatibility. Among the different ligands used to functionalize these particles, proteins, peptides, aptamers, and small molecules are the most widely employed to promote active BBB crossing and GBM targeting of nanosystems. 

Box 1Nanoparticle for non-invasive brain drug delivery.Nanoparticulate systems comprise a wide range of carriers such as lipid-based nanoparticles, polymer and dendrimer nanocarriers, metallic and inorganic nanoparticles. They differ in several features, like size, shape, porosity, chemical composition and surface properties (charge and functionality). Surface chemistry can affect not only the cellular uptake and the distribution of nanopartiles, but also their ability to cross biological barriers, like BBB, and the pharmacokinetics properties. Nanoparticles have been investigatd as drug delivery platforms for many years with the approval of the first nanoformulation for cancer therapy (Doxil^®^ and Marqibo^®^) more than 20 years ago. However, now there are few approved nanoparticle-based therapeutics and none for the treatment of brain disorders. The reason is the difficulty in the design of nanoparticles with an optimal combination of long half-life, BBB crossing and drug payload.

Some studies related to the ability of different cells to interchange nanoparticles through TNTs are available [130]. It has been shown that fluorescently labelled silica nanoparticles can be transferred between tumor cells by TNTs [131]. Franco S et al. demonstrated the intercellular trafficking of mesoporus silica nanoparticles along TNTs and TMs between macrophages and cancer cells in vitro and in vivo [132]. A recent paper from our group reported the exchange of multifunctional liposomes between human GBM cells and healthy astrocytes in vitro. Interestingly, the TNT-mediated transport of liposomes was more efficient between tumor cells compared with healthy astrocytes [123].

This highlights the structural differences in TNTs formed between tumor and healthy cells, which reflect a different rate of material exchange, which can be used to improve the precision of treatments. These findings support the exploitation of TNTs for cell-to-cell transfer of drug delivery systems to maximize treatment efficacy and efficiency.

### 6.3. Insights and TNTs Research Outlook

The adoption of TNTs as therapeutic or prognostic targets is an attractive approach. At least two alternatives have been proposed: (1) the identification of specific markers in order to design molecules that inhibit TNTs formation between cells in the TME and (2) the enhancement of drug delivery between cells by exploiting these channels. In this context, we foresee that TNTs are useful drug-delivery channels for cancer therapy, as they facilitate the intercellular distribution of the drug (or drug delivery systems) between close and distant cells that are not sufficiently targeted upon simple drug diffusion in the brain parenchyma. Although sustained by few experimental data thus far, this approach is innovative and intriguing and further investigation is warranted.

In the field of GBM treatment, GSCs represent a challenge due to their resistance to commonly employed anticancer drugs. Moreover, this issue is worsened by the localization of these cells in tumor niches, often far away from the tumor mass. Therefore, the combination of long TNTs formation from cancer cells that colonize normal tissue [133] and their intercellular transport ability makes their targeting a valuable approach for the prevention and treatment of GBM recurrence. In addition, the possibility of TNTs formation between different cell types, e.g., between tumor and immune cells, and the structural differences between TNTs formed by tumor vs. healthy cells could be exploited to boost the precision of nanocarrier delivery.

## 7. Conclusions

This review explores the heterogeneity of the GBM environment, with a focus on the intercellular communication between different cell types, which facilitates biological diversity within the tumor and rapidly evolving TME. We explain the emerging role of TNTs and outline the perceived current limitations in the field that must be addressed before pharmacological targeting of TNTs can become a clinical reality. A key requirement is the standardization of the terminology surrounding and definition of TNTs as well as the development of appropriate tools suitable for the detection and characterization of these structures in vivo. We conclude that TNTs provide a new and intriguing avenue to target the key cellular players in GBM with nanoparticles and their incorporated drugs.

## Figures and Tables

**Figure 1 cancers-13-04001-f001:**
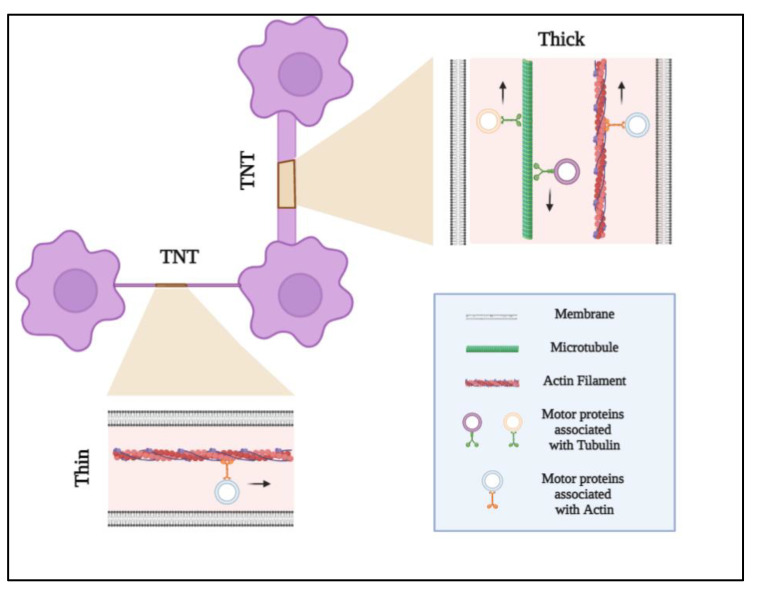
Tunneling nanotube (TNT) structures. TNTs are heterogeneous, transient, and highly dynamic nanotubular structures that connect cells over short or long distances. TNTs can be classified as two types: (1) short (100–200 nm), thin (≤0.7 μm), and dynamic type I nanotubes are made of actin and formed by cells exploring their surroundings to make cell–cell contact; (2) long (1 μm), thick (≥0.7 μm), and more stable type II nanotubes are made of tubulin and actin filaments. TNTs transport cellular organelles such as mitochondria and lysosomes, as well as viruses, intra-cellular vesicles, and electrical signals. The mechanism of cargo transport can be unidirectional or bidirectional. Type I nanotubes (actin-based) are characterized by a unidirectional transport mechanism in which cellular components are anchored and driven by the directional actin polymerization at one end. The hallmark of type II nanotubes is bidirectional delivery, in which cargo is moved by microtubule-based molecular motor proteins.

**Figure 2 cancers-13-04001-f002:**
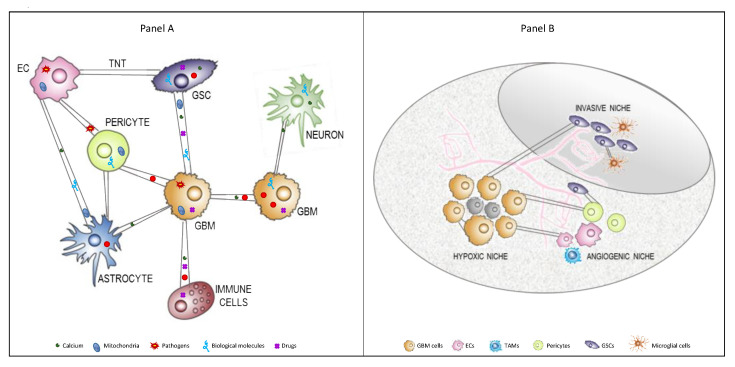
TNTs between cells composing the GBM TME. (**A**) TNTs are long membranous open-ended channels connecting cancer cells between them or with other cells composing the TME. They are dynamic and sensitive structures of different lengths and thicknesses able to allow for the interchange of organelles, vesicles, macromolecules, and pathogens between connected cells. (**B**) GBM and GSCs are embedded in a heterogeneous TME, which is composed of diverse stromal cells, including vascular cells, infiltrating and resident immune cells, and other non-neoplastic glial cell types, but they are also compartmentalized in distinct brain areas, called niches. These niches regulate metabolic needs, immune surveillance, survival, invasion, and the progression of GBM. TNTs are able to physically connect these cells even if located in different niches. Abbreviations: TNT, tunneling nanotube; TME, tumor microenvironment; EC, endothelial cells; GSCs, glioblastoma stem cells; GBM, glioblastoma; TAMs, tumor-associated macrophages.

**Figure 3 cancers-13-04001-f003:**
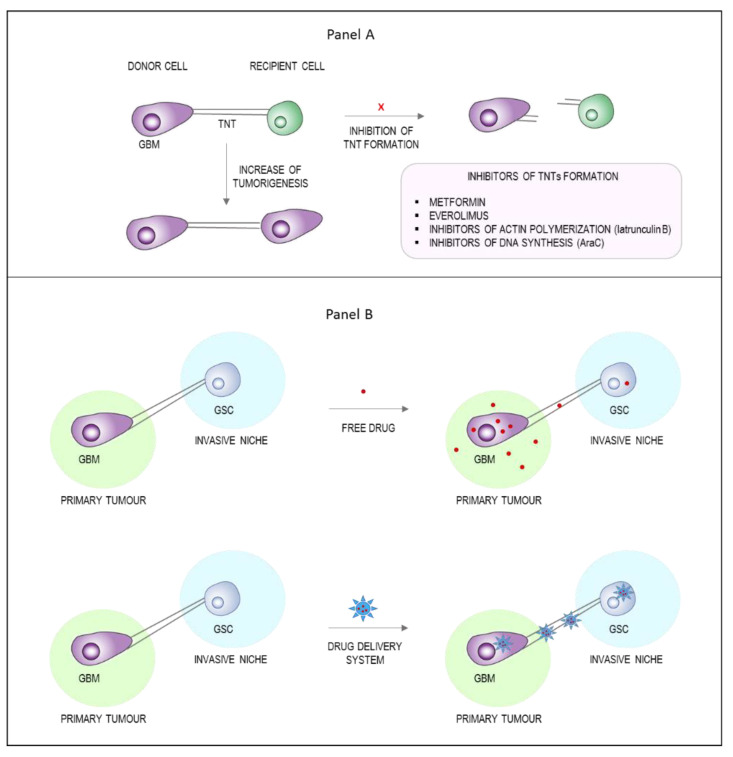
Potential therapeutic approaches based on TNTs. TNTs have been observed in several types of cancer, where they emerge to steer a more malignant phenotype thanks to the exchange of intracellular materials. (**A**) The design of molecules able to inhibit the formation of TNTs (i.e., inhibitors of actin polymerization, inhibitors of DNA synthesis, or inhibitors of mTOR pathway) may be a new therapeutic opportunity. (**B**) Since TNTs are long membrane nanotubes connecting distant cells, they can represent an exciting opportunity to deliver drugs or drug delivery systems in order to target inaccessible brain regions where GSCs are hidden and unreachable. Abbreviations: TNTs, tunneling nanotubes; GSCs, glioblastoma stem cells; GBM, glioblastoma.

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
