# Peer review of "The 3.0 Cell Communication: New Insights in the Usefulness of Tunneling Nanotubes for Glioblastoma Treatment"

_cancers, 2021, doi:10.3390/cancers13164001_

Round 1

Reviewer 1 Report

This review wants to present new insights into the formation, functions and possible therapeutic uses of tunnelling nanotubes (TNTs) in glioblastoma.

This is a growing area of research, since these poorly described structures might play crucial in the communication within the tumor microenvironment.

In its present form, the review might need some text adjustments and clarifications.

  • The authors cite high number of reviews, rather than original research articles, this should be changed. Seminal papers should be mentioned.
  • Section 4 (lines 213-72) is not well structured. It would ease the reading to articulate this section in sub-sections, like: i) starting with a general definition and the initial discovery of TNTs, ii) reiterating how they are thought to be formed, and iii) presenting their suspected functions and biological effects.
  • Likewise, section 4.1 is not clearly structured either and might need re-organization, to highlight the focus of the authors took on explaining TNTs in the immune responses.
  • Again, section 5 could be articulated in subsections to highlight the two possible therapeutic strategies.
  • In several places, text appears really vague and somehow too general. Precisions are needed in the following places: section 2.1 (immune infiltration within the CNS, glymphatic system), here the concepts are not clearly presented and seminal citations are missing – section 4 (general in vitro definition of TNTs) – section 4.1 lines 287-91 – section 4.1 lines 309-14 – section 4.1 lines 319-23 – section 5 lines 352-4 – section 5 lines 372-80 and 387-9.
  • The usual abbreviation for “GAMMs” is GAMs.
  • Paragraph on the BBB and drug delivery is rather naïve: section 5.1 – lines 415-23. Similar comments can be done on the paragraph describing the use of metformin and mTOR inhibitors (section 5, lines 356). Text clarifications are required. Box 2 is not relevant for the review.
  • Paragraph section 5, lines 407-14 is too speculative
  • Background on nanoparticles in therapies is necessary (section 5, lines 372).
  • Some sentences are difficult to understand and need further clarifications: last sentence of the abstract, lines 94-6, lines 108-11, lines 129-31, lines 168-70, lines 265-7, lines 282-6.
  • Line 128, GSCs may also give rise to endothelial cells in addition to pericytes.
  • The description of EVs needs to be improved (lines 187-92).
  • Please check meaning and edit text: lines 15 “in the distance”, 17 “to perform”, 20 “valid strategy”, 20 “one way or another”, line 25 “in charge of”
  • Lines 124, Is it clearly established that GSC niches and locations are based on an “anatomically” definition? Again, some essential concepts should be detailed in the section2.2, such as GBM subtypes (Verhaak et al. 2010).
  • Lines 324-6, this paragraph does not appear logically placed here.
  • Figures need to be improved with more details, the text in the top it is not necessary. A figure detailing TNT structure is important to reach a broader audience.
  • The explanation of miRNAs in EVs might appear rather too detailed and a bit out of the scope of the review (section 3 - lines 202-208).

Author Response

Point 1. The authors cite high number of reviews, rather than original research articles, this should be changed. Seminal papers should be mentioned.

Response 1. New references (accordingly to Reviewers’ suggestions) and seminal papers have been added or substituted to some cited reviews. References have been re-numbered, and the reference list modified.

Point 2. Section 4 (lines 213-72) is not well structured. It would ease the reading to articulate this section in sub-sections, like: i) starting with a general definition and the initial discovery of TNTs, ii) reiterating how they are thought to be formed, and iii) presenting their suspected functions and biological effects.

Response 2. Section 4 “Tunneling nanotubes” has been completely restructured and articulated in three subsections following the reviewer’s suggestions. Now, in the revised version, this section appears clearer and easier to read and understand by readers.

Point 3. Likewise, section 4.1 is not clearly structured either and might need re-organization, to highlight the focus of the authors took on explaining TNTs in the immune responses.

Response 3. Section 4.1 “Tunneling nanotubes in glioblastoma” have been reorganized and placed as a single section (now section 5 in revised manuscript) to highlight the focus of the review. This section has been divided into two subsections to clearly discuss: i) the identification and formation of TNTs in GBM between different cells composing the GBM microenvironment (now section 5.1 in revised version); and ii) the role and function of TNTs in GBM biology (now section 5.2 in revised version). Many sentences have been rewritten.   

Point 4. Again, section 5 could be articulated in subsections to highlight the two possible therapeutic strategies.

Response 4. Following the reviewer’s suggestions, the section 5 “TNTs as novel strategy to enhance tumor drug delivery” has been renumbered (now section 6 in revised version of the manuscript) and articulated in two sections based on the two possible therapeutic strategies exploiting TNTs: section 6.1 inhibition of TNTs formation and 6.2 exploiting TNTs as drug delivery channels. Many sentences have been rewritten.   

Point 5. In several places, text appears really vague and somehow too general. Precisions are needed in the following places: section 2.1 (immune infiltration within the CNS, glymphatic system), here the concepts are not clearly presented and seminal citations are missing – section 4 (general in vitro definition of TNTs) – section 4.1 lines 287-91 – section 4.1 lines 309-14 – section 4.1 lines 319-23 – section 5 lines 352-4 – section 5 lines 372-80 and 387-9.

Response 5. In section 2.1, concepts regarding the immune infiltration in the brain and the glymphatic system have been rewritten in a more precise way. Accordingly, some references have been substituted with seminal refs. In section 4 a more detailed description of TNTs and a new figure (now Figure 1 in revised version of the manuscript) about the TNTs structure have been added. Lines 287-91, 309-14, 319-23 of section 4.1 have been reorganized in a more precise form and shifted in a more appropriate section or subsection. All section 5 (now section 6 in revised version) has been rewritten and many sentences have been better articulated. 

Point 6. The usual abbreviation for “GAMMs” is GAMs.

Response 6. The text has been changed, accordingly.

Point 7. Paragraph on the BBB and drug delivery is rather naïve: section 5.1 – lines 415-23. Similar comments can be done on the paragraph describing the use of metformin and mTOR inhibitors (section 5, lines 356). Text clarifications are required.

Response 7. The reviewer’s comment on paragraph about the BBB and drug delivery is appropriate and extremely important for the focus of the manuscript. Accordingly, it has been removed for being out of the scope of the manuscript. Many thanks.

The paragraph describing the use of metformin and everolimus as TNTs inhibitors has been rewritten to better contextualize it in the section 6.1 of revised version of the manuscript.

Point 8. Box 2 is not relevant for the review.

Response 8. Box 2 has been removed accordingly to the reviewer’s suggestion.

Point 9. Paragraph section 5, lines 407-14 is too speculative.

Response 9. This paragraph has been rewritten trying to make it less speculative. 

Point 10. Background on nanoparticles in therapies is necessary (section 5, lines 372).

Response 10. A paragraph has been added on the base of reviewer’s comment (pp. 11 of revised version).

Point 11. Some sentences are difficult to understand and need further clarifications: last sentence of the abstract, lines 94-6, lines 108-11, lines 129-31, lines 168-70, lines 265-7, lines 282-6.

Response 11. All these sentences have been rewritten in a clearer way in the revised version of the manuscript.

Point 12. Line 128, GSCs may also give rise to endothelial cells in addition to pericytes.

Response 12. This concept has been added.

Point 13. The description of EVs needs to be improved (lines 187-92).

Response 13. The description of EVs has been improved (section 3, pp. 5 of revised version). Accordingly, some references have been rearranged or added to the text.

Point 14. Please check meaning and edit text: lines 15 “in the distance”, 17 “to perform”, 20 “valid strategy”, 20 “one way or another”, line 25 “in charge of”

Response 14. The text in these lines has been edited.

Point 15. Lines 124, Is it clearly established that GSC niches and locations are based on an “anatomically” definition?

Response 15. Many thanks for this clarification. Actually, GSC niches are not anatomically distinct brain regions, but they are specific and particular areas in glioblastoma microenvironment.

Point 16. Again, some essential concepts should be detailed in the section2.2, such as GBM subtypes (Verhaak et al. 2010).

Response 16. Details on GBM subtypes and the suggested reference (39) have been added in revised version of the paper.

Point 17. Lines 324-6, this paragraph does not appear logically placed here.

Response 17. The paragraph has been moved to section 2.1 of revised manuscript.

Point 18. Figures need to be improved with more details, the text in the top it is not necessary. A figure detailing TNT structure is important to reach a broader audience.

Response 18. Figures have been modified by adding more details, the text in the top has been removed and a new figure on TNT structure (Figure 1 of revised version) has been added. Accordingly, the figures have been renumbered and the figure legends modified.

Point 19. The explanation of miRNAs in EVs might appear rather too detailed and a bit out of the scope of the review (section 3 - lines 202-208).

Response 19. We completely agree with the reviewer’s point of view, therefore this section has been removed in the revised paper.

Reviewer 2 Report

The authors of this work discuss the role of tunneling nanotubes in the biology of glioblastoma and as potential therapeutic targets. Overall, the topic is highly interesting and certainly deserves a comprehensive review. However, regarding this manuscript I do have the following concerns which preclude publication at this point:

The language needs serious editing. There are a multitude of grammatical errors and typos. Some of the sentences need to be restructured (l.108-110; l. 206-208; l. 332-335; l. 384-385)

l. 49 glioblastoma has been assigned who grade IV

l. 56 the role of TTF as first-line therapy in newly-diagnosed glioblastoma should be mentioned

Author Response

Point 1. The language needs serious editing. There are a multitude of grammatical errors and typos. Some of the sentences need to be restructured (l.108-110; l. 206-208; l. 332-335; l. 384-385).

Response 1. The entire text has been undergone to an extensive English revision by a native English-speaking colleague (Dr. Alysia Cox). The grammatical errors and typos have been corrected in the revised version of the manuscript. The sentences indicated by the Reviewer, and others, have been completely restructured. In general, all the manuscript has been reorganized as suggested by Editor and Reviewers.

Point 2. L. 49 glioblastoma has been assigned who grade IV

Response 2. The sentence has been corrected accordingly.

Point 3. L. 56 the role of TTF as first-line therapy in newly-diagnosed glioblastoma should be mentioned

Response 3. The tumor treating field (TTF) as an adjuvant therapy for newly diagnosed glioblastoma has been added to the revised version of the manuscript. Consequently, a new reference (3) has been added.

Round 2

Reviewer 1 Report

The authors have done a really good job in answering my concerns and adequately revised their manuscript.

Reviewer 2 Report

The manuscript has substiantially improved.